# Effects of Foam Rolling vs. Manual Therapy in Patients with Tension-Type Headache: A Randomized Pilot Study

**DOI:** 10.3390/jcm11071778

**Published:** 2022-03-23

**Authors:** Gemma V. Espi-Lopez, Marta Ingles, Juan J. Carrasco-Fernandez, Pilar Serra-Añó, Luis Copete-Fajardo, Juan Jose Gonzalez-Gerez, Manuel Saavedra-Hernandez, Elena Marques-Sule

**Affiliations:** 1Department of Physical Therapy, Faculty of Physical Therapy, University of Valencia, 46010 Valencia, Spain; gemma.espi@uv.es (G.V.E.-L.); marta.ingles@uv.es (M.I.); juan.j.carrasco@uv.es (J.J.C.-F.); pilar.serra@uv.es (P.S.-A.); copete19@hotmail.es (L.C.-F.); elena.marques@uv.es (E.M.-S.); 2Department of Nursing, Physiotherapy and Medicine, Faculty of Health Sciences, University of Almería, c/Carretera de Sacramento s/n, La Cañada, 04120 Almería, Spain; juanjogg@ual.es; 3Fisiosur I + D Research Institute, Garrucha, 04630 Almería, Spain

**Keywords:** tension-type headache, physical therapy modalities, clinical trial

## Abstract

Background: This study compares the effect of foam rolling (FR) vs. manual therapy (MT) on pain, pressure pain threshold (PPT), headache disability (HDI) and impact of headache (HIT-6) in patients with tension-type headache (TTH). Methods: A total of 38 participants with TTH were randomly assigned to an FR group (FRG, *n* = 13), an MT group (MTG, *n* = 13) or a control group (CG, *n* = 12). FRG received FR treatment; MTG received MT techniques; CG received a placebo treatment. The treatment lasted one month. Outcome measures were assessed at baseline, post-intervention and follow-up. Results: Both FRG and MTG showed significant improvements in all variables after the treatment, but the intervention effect was maintained only for functional disability (*p* = 0.002 and *p* = 0.005, respectively), overall disability (*p* = 0.007; *p* = 0.030) and HIT-6 (*p* = 0.002; *p* = 0.001) at follow-up. After treatment, FRG and MTG presented a significantly higher PPT in right (*p* = 0.044; *p* = 0.009) and left suboccipital (*p* = 0.004; *p* = 0.021). MTG showed a significantly lower HIT-6 than CG (*p* = 0.008). No differences between FRG and MTG were found in any variable. Conclusions: Both FR and MT are effective treatments for the improvement of clinical symptoms in TTH. Further studies are needed to confirm our findings in a larger population.

## 1. Introduction

Tension-type headache (TTH) is a primary headache reclassified by the International Headache Society (IHS) in 2018, which describes in detail the diagnostic criteria for each type of headache. It is a typically bilateral headache that lasts from 30 min to 7 days. The diagnosis of TTH includes at least two of the following characteristics: bilateral location, pressing or tightening in quality mild or moderate intensity, and it does not worsen with routine physical activity [1]. The increased tenderness in the pericranial muscles is the most significant abnormal finding seen in patients with TTH. This tenderness typically occurs between attack, worsens during the headache and increases with intensity and frequency. Hypersensitivity seems to have a pathophysiologic importance [1]. TTH is mainly divided into episodic and chronic types. The infrequent episodic tension-type headache (IETTH) includes at least ten episodes of headache that appear on average less than one day a month. However, the frequent episodic tension-type headache (FETTH) appears with at least ten headache episodes that occur on average 1–14 days per month for more than three months and does not cause nausea or vomiting. On the other hand, in chronic tension-type headache (CTTH), the episodes occur with a mean ≥ 15 days per month for more than three months with the same duration as IETTH and FETTH, or even without remission [1].

Overall, TTH constitutes the most common primary headache with a prevalence throughout life ranging from 30% to 78% in the general population [1,2], a slightly higher incidence among women than men. The gender-adjusted 1-year prevalence of TTH is 38.2% [3]. This type of headache was considered primarily psychogenic; however, several studies strongly suggested the possibility of a neurobiological basis, at least for the most severe subtypes of TTH. The differences between episodic and chronic TTH that were presented in ICHD-I have been proven to be extremely useful to clinicians, especially in those cases where diagnosis is uncertain [1]. The IHS recommends comparing patients diagnosed according to each group of criteria, not only to categorize the clinical characteristics, but also to investigate the pathophysiological mechanisms and the response to treatments [1]. Several factors can trigger headache, such as emotional and genetic factors, as well as peripheral factors (pericranial tenderness, muscle strain, muscle blood flow and ischemia) or central factors, such as pain sensitivity, central pain modulation, etc. Although the disease burden is higher in CTTH than ETTH, both entities significantly affect the patient’s quality of life, thus having a high socio-economic impact [1,4].

Despite numerous clinical and neurophysiological studies, the underlying cause of TTH remains unclear. It has been suggested that recurrent nociceptive stimuli from pericranial myofascial tissues result in sensitization of pain pathways in the central nervous system [5]. Peripheral activation or sensitization of myofascial nociceptors is most likely involved in developing muscle pain and acute episodes of TTH. In turn, repetitive episodes of muscle pain can sensitize the central nervous system and lead to the progression of TTH to the chronic type [6]. Furthermore, it has been proposed that trigger points (TPs) may be a key feature of TTH. In this regard, active TPs in the upper trapezius, sternocleidomastoid and temporalis muscles have been associated with headache intensity and duration [7]. Thus, an updated pain model for TTH has been proposed, in which headache may be explained by referred pain from TPs in the posterior shoulder, head and cervical muscles, with TPs being the hyperalgesic areas responsible for the central sensitization [8].

Although drugs are considered as the first-line treatment, non-pharmacology treatments in TTH show positive results [9]. However, more studies are necessary to strengthen the evidence of non-pharmacology treatments in TTH [10], including manual therapy (MT) [11,12,13]. Despite the heterogeneity of studies, MT has been shown to induce positive effects on TTH-associated symptoms [9,14]. In particular, manual cranial therapy has been shown to reduce pain intensity in patients with TTH [15]. We previously reported that an MT treatment for TTH based on the combination of suboccipital soft tissue inhibition and occiput-atlas-axis manipulation improved several symptoms, including photophobia, phonophobia and pericranial tenderness [12]. This therapeutic approach also enhanced the quality of life in different dimensions related to moderate physical activities and the emotional role [13]. We further demonstrated that these gains were more significant when myofascial trigger point treatment was added [16]. We have also reported that larger increases in the upper cervical flexion range of motion can be observed when a set of manipulative techniques is added to a classic massage-based protocol [17].

In addition, a protocol combining MT-based techniques on the pelvis, skull, cervical and upper thoracic spine, clavicles and upper ribs has been proven to be effective in reducing TTH frequency [18]. However, a protocol including soft tissue techniques, TPR, occiput-atlas-axis manipulation and lumbar-sacral technique on TTH-related symptoms has not yet been investigated.

Foam rolling (FR) is a novel self-administered myofascial release technique that uses a foam roller (i.e., a cylinder covered with foam of different textures) to enhance myofascial mobility [19]. The pressure applied with the FR aims at provoking a reaction in the Golgi tendon organs, increasing vascularity and muscle temperature, thus relieving muscle rigidity and muscle spasm [20]. Indeed, FR reduces arterial stiffness and improves vascular endothelial function [21]. Existing evidence suggests FR can enhance joint range of motion (ROM) [22] and the recovery process by decreasing the effects of acute muscle soreness [23] and delayed onset muscle soreness (DOMS) [24] and improve muscle performance [25] in healthy patients. In the context of rehabilitation, it has been reported that FR enhances the quality of life in people with fibromyalgia. However, the effects of FR on TTH-related symptoms (i.e., pain, pressure pain threshold (PPT), headache disability and impact of headache) have not been explored.

We hypothesized that both MT and FR treatments applied to patients with TTH would lead to significant improvements in different variables related to headache. Hence, the objective of the present study was to assess the effectiveness of two exercise programs based on FR and MT, respectively, on pain, PPT, headache disability and impact of headache in patients with TTH.

## 2. Materials and Methods

### 2.1. Participants

Participants aged 20–55 years and diagnosed with FETTH (at least four episodes/month) or CTTH by a primary care physician according to “The International Classification of Headache Disorders, 3rd edition” (ICHD-3) [1] were recruited from a rehabilitation clinic from August to December 2018. All enrolled participants provided written informed consent before participation. Detailed inclusion and exclusion criteria are presented in Table 1. 

### 2.2. Study Design

The study was a prospective, assessor-blinded, randomized pilot study. Level of Evidence: Therapy, level 2. 

### 2.3. Randomization and Masking

Randomization was undertaken by an independent research assistant not involved in the trial, who prepared a computer-generated random allocation sequence. From the randomization allocation, opaque, sequentially numbered envelopes were ready to contain the treatment group assigned for each participant. Thus, patients were randomly assigned to three groups: (1) FRG, (2) MTG and (3) Control group (CG). Group allocation was revealed to the study members once the participants completed all baseline procedures. The outcomes assessor was blinded to treatment group allocation. The physiotherapist was not informed of the type of study being conducted or the aim. Participants were reminded not to reveal their treatment group at the post-intervention examination. Blinded assessors collected all baseline and post-intervention measures and entered data. All patients were assessed before treatment, after treatment (at four weeks) and at a follow-up (one month after completion of therapy).

### 2.4. Intervention

The treatment was composed of four sessions at 7-day intervals for all groups. Treatments were performed at a university research laboratory. Interventions were delivered by the same physiotherapist experienced in the trial treatments. Before each treatment session, the physiotherapist performed the vertebral artery test bilaterally in all patients. The vertebral artery test has been widely used as one of the methods of screening the vertebro-basilar system prior to manipulative therapy [26]. The occurrence of adverse events was monitored throughout and between the treatment sessions. 

(a)MTG. A protocol composed of six MT techniques was applied. In total, each MT session lasted 40 min. After MT treatment, all groups rested for 10 min in a supine position with neutral ranges of neck flexion, extension, lateral flexion and rotation [13,27]. The treatment consisted of:
(1)Compression of the fourth ventricle (CV-4). The participant lay down in a supine position. The physiotherapist, situated behind the participant’s head, slightly approximated the occipital squama lateral angles toward the posterior occipital convexity while taking the cranium into extension. The compression traction was maintained until an inactive state was perceived in the cranial pulse and released when the perception of movement was noted. The duration of this technique was 10 min [15].(2)Suboccipital inhibitory pressure: suboccipital musculature was palpated until contact was made with the posterior arch of the atlas, and progressive and deep gliding pressure was applied, pushing the atlas anteriorly. The occiput rested on the hands of the physiotherapist while fingertips supported the atlas. Finger pressure was maintained for 10 min to produce the proposed therapeutic effect of inhibiting the suboccipital soft tissues [12].(3)Trigger point release (TPR) was applied bilaterally in the trapezius to relax the trapezius, since tension may influence the occipital region due to the trapezius insertion. Myofascial trigger points were identified using published criteria [28]. Trapezius was palpated for a tender nodule along with taut bands. Force was progressively applied to the nodule, with the patient instructed to verbally indicate whether they felt pain locally or referred pain. The physiotherapist applied a pincer grip of sufficient force to elicit referred pain (or 6 on a 10-point scale) to the identified site. The duration was until the participant verbally reported dissipation of referred pain, the physiotherapist detected a physical softening in the trigger points, or a maximum of 60 s had elapsed. Up to five compressions were performed at each site with a 10 s rest between contractions [16]. In total, this technique lasted 6 min.(4)Occiput-atlas-axis technique. This was applied bilaterally and performed on a vertical axis, passing through the odontoid process of the axis without extension or flexion and very little side bending. The technique was used in two stages. First, the physiotherapist performed a light decompression and then made a small circumduction. Then, the appropriate joint barrier was sought by selective tension, and high-velocity rotation manipulation was performed in a cranial helical motion without raising the participant’s head [15,17]. This technique lasted 2 min. All of these manual osteopathic treatments are believed to improve circulation, release joint restrictions, reduce tension in muscles, fascia and dura, decrease nociceptive input and promote a normalizing or calming effect on the CNS according to Hanten et al. [15].(5)Lumbar-sacral technique. Since the relationship between a lack of mobility of cranial bones, sacrum and iliac bones, and TTH has been reported [18], three lumbar-sacral mobilizations were performed, and the appropriate joint barrier was sought by selective tension. In lateral decubitus, the patient placed the lower leg with knee extension, the upper portion with hip and knee flexion, setting the foot at the level of the popliteal fossa of the opposite leg. Then, a trunk rotation manipulation was performed, and the appropriate joint barrier was sought by selective tension. The technique was performed bilaterally and lasted 2 min.(6)Cranial massage: classic massage techniques were applied based on the study conducted by Weerapong et al. [29]. Methods used included soft and shallow effleurage and digital petrissage, and tapotement, a superficial massage composed of gentle rubbing and kneading, was performed in the skull and cervical region for 10 min (5 min prone and 5 min supine).

(b)FRG. Participants were given a brief introduction to the FR procedure. A foam roller with a length of 38 cm and a diameter of 12.5 cm was used (Domyos, Paris, France). An exercise protocol focused on the back’s fascia (suboccipital, cervical and sacral region) was developed [30]. Participants performed four FR exercises under supervision. All exercises were repeated for 1 min, rested for 30 s, and then the procedure was repeated for another minute. Participants rolled out the muscles three times during each minute of FR [31]. Each technique lasted 6 min. In total, the FR treatment lasted 25 min.The rolling frequency was standardized using a metronome set at 60 beats per minute (bpm). Participants were instructed to roll at a velocity of 2 metronome beats (thus, 2 s) for each rolling direction, resulting in 15 complete rolling cycles in 60 s (0.25 Hz). The intensity of pressure was controlled with a target Numerical Rating Scale rating of 7/10 (0 = no discomfort, 10 = maximal discomfort) during the intervention. The treatment was as follows (see Appendix A):
(1)Middle-upper trapezius and interscapular muscles: participants were instructed to roll the FR up using flexion extension in the supine position, with the FR at the interscapular muscles until the FR was just above the upper trapezius. Then, participants were told to roll the FR back to the initial position in one fluid motion.(2)Suboccipital muscles: in the supine position, with the FR at the suboccipital region, suboccipital flexion was required until the FR was just above the upper trapezius. Then, participants were told to roll the FR back to the initial position.(3)Posterior and lateral cervical muscles: in the supine position, with the FR at the posterior cervical muscles, participants were asked to rotate the neck and then roll the FR back to the initial work, bilaterally.(4)Sacrum region: participants were in the supine position, with the FR at the proximal part of the sacrum and their body supported by the hands placed behind them. From this position, they were instructed to roll the FR down using flexion extension of the knees until the foam roller was just above the distal region of the sacrum. Then, participants were told to move the FR back to the initial position.

(c)CG. In the supine position, a series of short-time and no pressure contacts with the physiotherapist’s hands was performed at several points on the head and shoulders for 10 min [32]. Then, patients remained 10 min in a resting supine position [13].

### 2.5. Outcome Measures

A baseline clinical interview was conducted to gather demographic and clinical information. Additionally, headache-related data were self-reported by the subjects in this interview, including aspects such as the evolution of headache, frequency, severity, intensity, pain profile, associated factors and pericranial sensitivity. Medication overuse remains a key factor in the interpretation, although in our sample there was no medication overuse.

#### 2.5.1. Primary Outcome

(a) Pain. A visual analog scale (VAS) was used. The VAS is an 11-point rating scale (0 = no pain to 10 = intense pain), indicating an average of TTH-induced pain in the last month. The VAS has a high internal consistency (0.92) [26,27,33].

#### 2.5.2. Secondary Outcomes

(b) Pressure pain threshold (PPT), i.e., the pressure at which a sensation of pressure changed to pain, was bilaterally assessed with a digital algometer (Somedic AB1, Somedic, Hörby (Farsta), Stockholm, Sweden) over the suboccipital muscle (distal to its origin and 2 cm from the midline) to determine overall pressure pain sensitivity. The pressure was applied perpendicular to the skin at a speed of 30 kPa/s. Three measurements with 1 min between measurements were averaged. This test has shown a high internal consistency (0.94–0.98) [34].

(c) Headache disability. We used the Headache Disability Inventory (HDI). The purpose of the scale is to identify the difficulties the patient may experience due to headaches. It includes two items: headache severity (mild, moderate and severe) and frequency (1/month; >1/month and <4/month; once/week) and 25 items that assess two subscales (emotional, 13 items; functional, 12 items). Participants answer each question (yes = 4 points, sometimes = 2 points, or no = 0 points). The maximum disability score is 100 points [35]. 

(d) Impact of headache, measured by the Headache Impact Test-6 (HIT-6) [36], which consists of 6 items, each with four response options: never, 6 points; rarely, 8 points; sometimes, 10 points; very often, 11 points; always, 13 points, with a total score ranging from 36 to 78 points. It has been shown to have a good internal consistency (Cronbach alpha 0.89) and test–retest reliability (ICC ranging from 0.78 to 0.90) [37].

### 2.6. Sample Size

An a priori power analysis was conducted in G*power (3.1.9.2 version) software to calculate the required sample size, using, as reference, the results obtained from published research with a similar approach. In previous studies, large effect sizes were obtained in all four HDI subscales (f = 1.22 or d = 2.44) and in HIT-6 scoring (r = 0.85 or d = 3.23) [15,16]. Therefore, with the current study design, accepting a 5% alpha risk (α = 0.05) as well as a 20% beta risk (β = 0.2), a total of 36 participants were required to achieve at least a medium effect size (f = 0.25 or d = 0.5).

### 2.7. Statistical Analysis

All statistical analyses were performed using the IBM SPSS Statistics software (Version 22.0; IBM Corp, Armonk, NY, USA). The Shapiro–Wilk test was used to evaluate the normality of the data. Descriptive data are shown as mean (standard deviation) or frequencies, as appropriate. Participants’ characteristics were compared using one-factor analysis of variance (ANOVA) or chi-square (χ^2^). 

A mixed 2-factor ANOVA with repeated measures in the time factor was used to determine significant differences in the measured variables between time points (pre-, post-treatment and follow-up) and groups (FRG, MTG and CG). When models indicated significant differences in the main effects, a Bonferroni correction was applied to avoid type I error in the multiple comparisons. The effect of the treatment in the HDI frequency and HDI severity variables was evaluated using χ^2^. Effect size was interpreted as small (d = 0.2; η_p_^2^ = 0.01), medium (d = 0.5; η_p_^2^ = 0.06) and large (d > 0.8; η_p_^2^ > 0.14). Statistical significance was set at *p* < 0.05.

### 2.8. Role of the Funding Source

The funders played no role in the design, conduct, or reporting of this study.

## 3. Results

Thirty-nine subjects were eligible, but one was excluded because of suffering from migraine. Then, 38 participants were randomly assigned to FRG (*n* = 13), MTG (*n* = 13) and CG (*n* = 12). All participants completed the study and were ultimately analyzed (Figure 1). No significant adverse effects were reported. Demographic data and clinical characteristics of the participants are depicted in Table 2. No between-group significant differences were found for any of the baseline variables. A total of 17.9% of the subjects had a family history of headaches in the sample. In the last month before the interview, TTH had a moderate impact on daily life, leisure and work activities in 25.0%, 28.6% and 22.3% of the subjects, respectively. It was only for 3.6%, 2.7% and 7.1% of participants that the impact of TTH on these activities was severe. Twenty-nine were women (76.3%), and nine were men (23.7%), with a mean age of 35 (13.46) years. None of the analyzed variables presented missing data.

The mixed-models ANOVA revealed significant differences in the time factor for all variables (Table 3). No significant differences were found in the group factor in any variable. Significant interaction group × time was observed in the VAS index, right and left suboccipital PPT and HIT-6 score. 

Table 4 and Table 5 show means (standard deviations) and post hoc results. As expected, no between-group differences were found at pre-treatment time in any variable. Although the VAS scale was lower in both intervention groups than in CG, no between-group differences were found post-treatment. However, at follow-up time, MTG showed a significantly lower VAS index than CG. FRG and MTG presented significantly higher PPT than CG, and the HIT-6 score was substantially lower in MTG than CG. In all cases, effect sizes were large (d > 0.8). No between-group differences were found in HDI items.

Regarding within-group analysis, both intervention groups showed a significant improvement in VAS scale, PPT, HDI items and HIT-6 scores after treatment. Effect sizes were large for VAS, MTG in the right suboccipital algometry, HDI item scores (medium for FRG in emotional HDI) and HIT-6. In the rest of the PPT measures, effect sizes were close to medium.

No significant differences were found in the FRG vs. MTG analysis, indicating that both treatments presented similar positive effects in the analyzed variables. For CG, pre–post significant differences were found in emotional, functional and overall HDI and HIT-6 scores, but with a medium effect size. Lastly, compared with the pre-treatment, at follow-up, the intervention effects remained significant in functional and overall HDI items for FRG and MTG and in all groups for HIT-6 scores.

The results of χ^2^ analysis (Table 6) showed that FRG and MTG, but not CG, presented a significant relationship between frequency of TTH and measurement time. We found differences in the evolution of headache in years and in the severity of the disorder. The control group presented more years of evolution of headache and more severe headaches than FRG and MTG. Nevertheless, these differences may be due to the randomization of the sample. Regarding HDI severity, none of the groups showed a meaningful relationship between the severity of the TTH and time of measurement. 

## 4. Discussion

In this study, both FRG and MTG showed a between-group significant improvement in PPT in the right and left suboccipital muscles post-intervention compared to CG. In addition, the MTG showed a considerable improvement in pain at follow-up and in the impact of headache in post-treatment compared to the CG. In the rest of the variables, there were no significant between-group differences.

On the other hand, concerning within-group comparisons, both FRG and MTG improved all variables in post-treatment, while the CG improved the impact of headache and the headache disability.

To the best of our knowledge, this is the first study comparing a protocol of FR with a protocol including several techniques of MT for the treatment of TTH. Thus, there is no literature regarding the potential effects of FR in TTH-related symptoms.

On the other hand, both FRG and MTG presented significant improvements and similar effects in all variables after treatment, further improvements in functional disability and overall disability, and the impact of headache at follow-up. 

Regarding the primary outcome, MT may be considered an effective treatment for pain in patients with TTH. In the current study, we observed short-term positive results both in FRG (−15.7%) and MTG (−20.8%); thus, both groups overcame the minimal clinically significant difference (MCID) (i.e., 15%) [38]. Similarly, it has been reported that TPR-, soft-tissue- and neural-mobilization-based MT treatments effectively reduce pain [39,40]. It has been reported that TTH induces the activation of trigger points, thus TPR seems to be recommended in these patients [7], and therefore, we included this technique in our MT protocol. Other studies using MT techniques found a decrease in headache pain compared to CG [14], as occurred in our study at follow-up, or by contrast, did not show a significant improvement [17].

With regard to PPT, we observed that FRG and MTG significantly improved this variable in both right (FRG: +34.8%, +0.33 kg/cm^2^; MTG: +43.3%. +0.67 kg/cm^2^) and left (FRG: +43.2%, +0.32 kg/cm^2^; MTG: +35.5%, +0.27 kg/cm^2^) suboccipital muscles when compared to CG. However, this outcome did not reach the MCID (1.5 kg/cm^2^) [41]. This is in agreement with several authors reporting positive effects of TPR-, soft-tissue- and neural-mobilization-based MT techniques on the cervical musculature, sternocleidomastoid muscle, temporal muscles and supraorbital region PPTs [38,42]. It should be taken into account that, in our study, absolute scores of PPT in FRG and MTG significantly improved when compared to CG, although the MCID was not achieved. 

This may demonstrate the importance of validating results based on absolute differences in outcomes scores against a measure of clinical relevance. 

As stated before, the effectiveness of FR as a valid treatment for pain relief has not been widely studied. Still, its beneficial effects on joint ROM, post-exercise muscle recovery, muscle soreness, muscle performance and PPT have been reported, especially in the sports field [18,23]. However, there is no consensus on the optimal FR dosage (treatment time, pressure and cadence) [18]. 

Our novel FR protocol showed short-term beneficial effects in PPT bilaterally in patients with TTH. Currently, no other studies are demonstrating the impact of FR in patients with TTH. A recent study reported beneficial effects of a protocol including FR, mobility and stretching exercises on ROM, frequency and pain intensity, fatigue, stiffness, depression/sadness and the disease’s impact on people with fibromyalgia [43]. On the one hand, our study is novel because previous papers investigating the effectiveness of MT in TTH [11,14,16,17,26,41] have used isolated MT techniques and measured a few variables. On the other hand, this is the first study to evaluate the effect of FR on TTH-related symptoms. Thus, MT and FR protocols seem to be proper tools with short-term effects for some TTH-related variables. 

Regarding disability, Moraska et al. assessed a 6-week suboccipital TPR-based MT protocol and obtained positive short-term and long-term results in overall and functional subscales [42]. Moreover, we previously reported that suboccipital inhibition and occiput-atlas-axis articulatory techniques [11,16] improved disability in TTH. Interestingly, we observed that both FRG and MTG reduced the frequency of TTH and the impact of headache (post-, FRG: −12.0%; MTG: −17.0%; follow-up, FRG: −9.0%; MTG: −9.3%). Then, both groups reached the MCID (i.e., 10%) after the treatment [44]. A short-term improvement in the latter has been reported by Moraska et al. [42] when applying a TPR-based MT. Thus, these results are coincident with those obtained in our study. Since the MCID is a frequently used threshold for changes in outcomes that can be of additional utility in the determination of the recovery process, we highlight that in the present study, both FGR and MTG achieved the MCID at the follow-up on functional disability, overall disability, frequency of TTH and impact of headache. 

Finally, we observed an effect of the placebo technique in some of the variables studied. We observed an effect of the placebo technique in some of the studied variables, specially referred to pre–post significant differences observed in emotional, functional and overall HDI and HIT- 6 score for CG, which might point to a placebo effect. We observe that, globally, these improvements were smaller (smaller effect size) than in FRG and MTG. This placebo effect could be due to the fact that the follow-up was carried out in a very limited period of time (one month). We believe that if the follow-up period had been longer (two or three months), perhaps we would not have observed any placebo effect. This aspect should be considered in future research. Although the placebo technique was not intended to treat, the application of manual contact may have improved some of the variables. In this regard, Kaptchuk et al. stated that placebo techniques could produce statistically and clinically significant results, with the patient–therapist relationship being the most robust component [45]. Other authors have suggested this may result from a psychological change in the patient, caused by the attribution of signification to treatment [46,47], as may have occurred in our study. 

### Strengths and Limitations

This is the first study that evaluates the effectiveness of FR on TTH from a therapeutic point of view. Our FR protocol allows health professionals to offer greater independence to patients regarding the management of TTH and an economical and accessible alternative. In addition, our protocol includes several MT techniques that have been proven to be effective separately but not together as a treatment protocol. This novel technique could be implemented in physiotherapy treatment, considering that it could be performed in person with physiotherapist supervision or as domiciliary treatment, as long as there are no contraindications for the administration. It is essential to highlight that in all cases, the FR technique should be supervised by a physiotherapist, as in the present study, to avoid self-administration errors. In this regard, the therapist should act as a guide, leading the correct way to perform the technique, while different in-person follow-up sessions should be conducted, and thus errors should be fixed. We consider that the use of both treatments may lead to beneficial effects and favor the implication and active collaboration of the participant. However, this study has limitations, such as the small sample size, although as a pilot study, it could be sufficient according to a priori power analysis. In addition, headache frequency was not registered with days of headache, which could explain the differences in results. Thus, any generalization should be made cautiously. Finally, scarce literature about the therapeutic approach of FR constitutes a difficulty when comparing the results with other studies. In this regard, further studies to confirm our findings in a larger population are needed.

## 5. Conclusions

The results of this study suggest that both FR and MT are effective treatments to improve pressure pain sensitivity in the short term in patients with TTH, with MT showing further long-term benefits on pain. Further studies with a larger sample and a more precise diagnosis of TTH are needed to confirm these results.

## Figures and Tables

**Figure 1 jcm-11-01778-f001:**
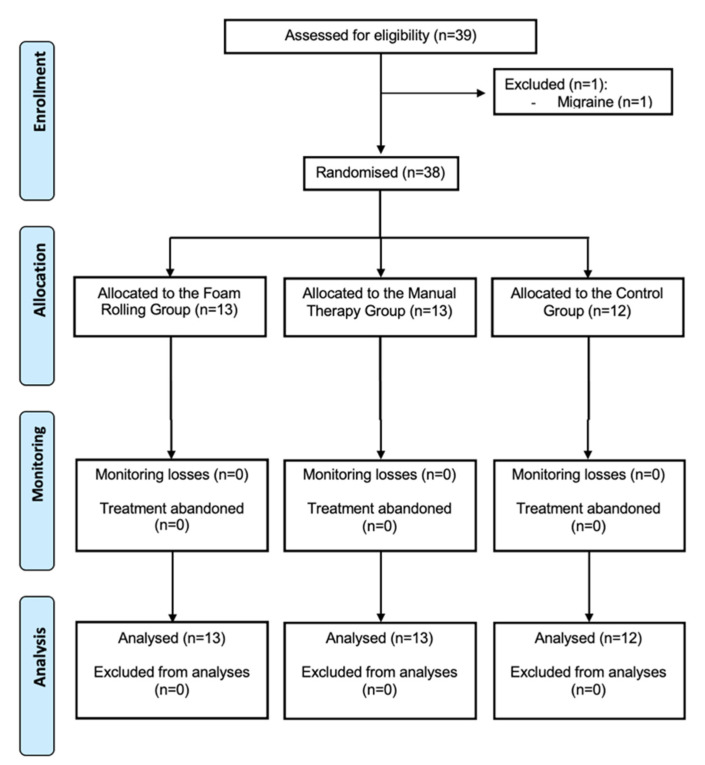
Flow chart according to CONSORT statement for the report of randomized trials.

**Table 1 jcm-11-01778-t001:** Inclusion and exclusion criteria.

Inclusion Criteria	Exclusion Criteria
- Aged 20–55 years - Medical diagnosis of FETTH or CTTH according to ICHD-3 - At least 3 months history of TTH - At least 4 days with headache per month	- Medical diagnosis of IETTH or any other headache disorder different from FETTH or CTTH according to ICHD-3 - Cervical trauma - Positive vertebral artery test - Contraindications to receiving manual therapy - No medication overuse - No medication preventive

FETTH: Frequent episodic tension-type headache. IETTH: Infrequent episodic tension-type headache. CTTH: Chronic tension-type headache. ICHD-3: The International Classification of Headache Disorders, 3rd edition.

**Table 2 jcm-11-01778-t002:** Baseline comparisons of demographic and clinical data.

	FRG (*n* = 13)	MTG (*n* = 13)	CG (*n* = 12)	Differences among Groups
Demographic variables				
Age (years)	31.08 (9.75)	29.62 (10.08)	35.58 (13.46)	F(2,35) = 0.96; *p* = 0.39
BMI (kg/m^2^)	25.28 (3.61)	25.50 (3.87)	28.50 (5.30)	F(2,35) = 2.16; *p* = 0.13
Gender, *n* (M/F)	3/10	4/9	2/10	χ^2^(2)= 0.69; *p* = 0.71
History				
Evolution of headache (years)	7.85 (7.94)	8.00 (7.23)	15.92 (12.16)	F(2,35) = 3.05; *p* = 0.06
Frequency, *n* (FETTH/CTTH)	9/4	9/4	8/4	χ^2^(2) = 0.03; *p* = 0.99
Severity of disorder, *n* (mild/moderate/severe)	1/12/0	0/11/2	0/7/5	χ^2^(4) = 8.90; *p* = 0.06
Intensity (VAS)	6.38 (0.65)	5.92 (1.50)	5.92 (1.31)	F(2,35) = 0.64; *p* = 0.54
Pain profile (triggers)				
Location of pain, *n* (front/parietal/occipital pain)	4/5/4	8/2/3	6/3/3	χ^2^(4) = 2.81; *p* = 0.59
Location bilateral pain, *n* (no/yes)	4/9	3/10	4/8	χ^2^(2) = 0.35; *p* = 0.84
Pressing or tightening (non-pulsating), *n* (no/yes)	3/10	6/7	4/8	χ^2^(2) = 1.54; *p* = 0.46
No pain increases with physical activity, *n* (no/yes)	9/4	10/3	7/5	χ^2^(2) = 1.00; *p* = 0.61
Time when the pain begins, *n* (morning/day/night/indifferent)	1/5/4/3	4/4/4/1	2/6/2/2	χ^2^(6) = 4.10; *p* = 0.66
Associated factors				
Photophobia, *n* (no/yes)	6/7	6/7	3/9	χ^2^(2) = 1.54; *p* = 0.46
Pericranial sensitivity, *n* (no/yes)	9/4	7/6	6/6	χ^2^(2) = 1.08; *p* = 0.58

Data are expressed as mean (standard deviation) or number of cases (*n*) as indicated. FRG: Foam rolling group; MTG: Manual therapy group; CG: Control group; BMI: Body max index; M: Male; F: Female; FETTH: Episodic tension-type headache; CTTH: Chronic tension-type headache; VAS: Visual analog scale. F and *p*-values were calculated using ANOVAs for continuous variables and Pearson’s χ^2^ tests for nominal or ordinal variables.

**Table 3 jcm-11-01778-t003:** Results of the ANOVA main effects for the analyzed variables.

	Time Factor	Interaction Group × Time
Pain (VAS)	*p* = 0.005; η_p_^2^ = 0.14	*p* = 0.047; η_p_^2^ = 0.13
Pressure pain threshold, right suboccipital (algometry, kg/cm^2^)	*p* = 0.005; η_p_^2^ = 0.14	*p* < 0.001; η_p_^2^ = 0.30
Pressure pain threshold, left suboccipital (algometry, kg/cm^2^)	*p* = 0.045; η_p_^2^ = 0.09	*p* = 0.002; η_p_^2^ = 0.21)
Emotional disability (emotional HDI)	*p* < 0.001; η_p_^2^ = 0.30	*
Functional disability (functional HDI)	*p* < 0.001; η_p_^2^ = 0.43	*
Overall disability (total HDI)	*p* < 0.001; η_p_^2^ = 0.41	*
Impact of headache (HIT-6)	*p* < 0.001; η_p_^2^ = 0.55	*p* = 0.010; η_p_^2^ = 0.17

VAS: Visual analog scale; HDI: Headache disability inventory; HIT-6: Headache impact test-6. No significant differences were found in the group factor in any variable. * No significant interaction.

**Table 4 jcm-11-01778-t004:** Results of the multiple comparisons for the analyzed variables.

	Pre-Treatment	Post-Treatment	Follow-Up	Pre vs. Post *p* [95% CI]; d	Pre vs. Follow-Up *p* [95% CI]; d
**Pain (VAS)**					
FRG (*n* = 13)	6.38 (0.65)	5.38 (1.19)	6.00 (0.91)	**0.010** [0.20:1.80]; 1.00	*
MTG (*n* = 13)	5.92 (1.50)	4.69 (0.86)	5.08 (1.75)	**0.001** [0.43:2.03]; 0.97	*
CG (*n* = 12)	5.92 (1.31)	5.92 (1.62)	6.83 (2.04)	*	*
FRG vs. MTG	*	*	*		
FRG vs. CG	*	*	*		
MTG vs. CG	*	*	**0.032** [−3.39:−0.12]; 0.90		
**Pressure pain threshold, right suboccipital (algometry, kg/cm^2^)**					
FRG (*n* = 13)	2.19 (0.81)	2.52 (0.62)	2.42 (0.66)	**0.045** [−0.66:−0.01]; 0.44	*
MTG (*n* = 13)	2.01 (0.69)	2.68 (0.74)	2.17 (0.35)	<**0.001** [−0.99:−0.35]; 0.90	*
CG (*n* = 12)	2.13 (0.79)	1.87 (0.53)	2.03 (0.58)	*	*
FRG vs. MTG	*	*	*		
FRG vs. CG	*	**0.044** [0.01:1.30]; 1.10	*		
MTG vs. CG	*	**0.009** [0.17:1.45]; 1.21	*		
**Pressure pain threshold, left suboccipital (algometry, kg/cm^2^)**					
FRG (*n* = 13)	2.30 (0.72)	2.62 (0.64)	2.47 (0.60)	**0.015** [−0.60:−0.05]; 0.45	*
MTG (*n* = 13)	2.19 (0.58)	2.48 (0.58)	2.40 (0.34)	**0.032** [−0.56:−0.02]; 0.48	*
CG (*n* = 12)	2.07 (0.63)	1.83 (0.47)	2.10 (0.58)	*	*
FRG vs. MTG	*	*	*		
FRG vs. CG	*	**0.004** [0.22:1.36]; 1.35	*		
MTG vs. CG	*	**0.021** [0.08:1.22]; 1.20	*		

Data are expressed as mean (standard deviation). Significant differences are highlighted in bold. * No significant differences. FRG: Foam rolling group; MTG: Manual therapy group; CG: Control group. d: effect size with Cohen’s d.

**Table 5 jcm-11-01778-t005:** Results of the multiple comparisons for HIT 6 and HDI.

	Pre-Treatment	Post-Treatment	Follow-Up	Pre vs. Post *p* [95% CI]; d	Pre vs. Follow-Up *p* [95% CI]; d
**Impact of headache (HIT 6)**					
FRG (*n* = 13)	63.31 (3.50)	55.85 (5.08)	57.62 (6.15)	<**0.001** [4.05:10.87]; 1.64	**0.002** [1.93:9.46]; 1.09
MTG (*n* = 13)	63.23 (3.61)	52.38 (7.43)	57.38 (7.19)	<**0.001** [7.44:14.25]; 1.78	**0.001** [2.08:9.61]; 0.98
CG (*n* = 12)	64.67 (4.03)	60.58 (6.23)	58.42 (6.60)	**0.019** [0.54:7.63]; 0.74	**0.001** [2.33:10.17]; 1.09
FRG vs. MTG	*	*	*		
FRG vs. CG	*	*	*		
MTG vs. CG	*	**0.008** [−14.56:−1.83]; 1.15	*		
**Functional disability (functional HDI)**					
FRG (*n* = 13)	26.62 (8.22)	17.23 (8.58)	17.62 (9.43)	<**0.001** [4.79:13.98]; 1.07	**0.002** [3.02:14.98]; 0.97
MTG (*n* = 13)	25.38 (9.71)	14.77 (7.85)	17.23 (9.33)	<**0.001** [6.02:15.21]; 1.15	**0.005** [2.17:14.14]; 0.82
CG (*n* = 12)	27.33 (7.30)	21.67 (9.57)	21.17 (12.61)	**0.016** [0.89:10.45]; 0.63	*
FRG vs. MTG	*	*	*		
FRG vs. CG	*	*	*		
MTG vs. CG	*	*	*		
**Emotional disability (emotional HDI)**					
FRG (*n* = 13)	17.54 (10.11)	11.85 (9.07)	11.08 (8.78)	**0.007** [1.33:10.05]; 0.57	*
MTG (*n* = 13)	17.69 (7.78)	9.69 (6.47)	13.08 (8.97)	<**0.001** [3.64:12.36]; 1.07	*
CG (*n* = 12)	23.50 (10.89)	17.50 (9.27)	19.17 (12.66)	**0.006** [1.46:10.54]; 0.57	*
FRG vs. MTG	*	*	*		
FRG vs. CG	*	*	*		
MTG vs. CG	*	*	*		
**Overall disability (total HDI)**					
FRG (*n* = 13)	44.15 (17.60)	29.08 (16.71)	28.69 (17.56)	<**0.001** [7.29:22.86]; 0.84	**0.007** [3.70:27.22]; 0.84
MTG (*n* = 13)	43.08 (16.94)	24.46 (12.76)	30.31 (17.34)	<**0.001** [10.83:26.40]; 1.19	**0.030** [1.01:24.53]; 0.71
CG (*n* = 12)	50.83 (17.28)	39.17 (17.13)	40.33 (24.63)	**0.003** [3.57:19.77]; 0.65	*
FRG vs. MTG	*	*	*		
FRG vs. CG	*	*	*		
MTG vs. CG	*	*	*		

Data are expressed as mean (standard deviation). Significant differences are highlighted in bold. * No significant differences. FRG: Foam rolling group; MTG: Manual therapy group; CG: Control group. d: effect size with Cohen’s d.

**Table 6 jcm-11-01778-t006:** Effects of the treatments on frequency and severity of headache disability.

		Frequency (HDI)			Severity (HDI)		
		<1 Episode/Month	1 to 4 Episodes/Month	>1 Episode/Week	Mild	Moderate	Severe
**FRG**	Pre-treatment	0	7	6	1	11	1
	Post-treatment	2	11	0	7	5	1
	Follow-up	1	11	1	5	7	1
	χ^2^ analysis	χ^2^ (4) = 12.0; *p* = **0.011**			χ^2^ (4) =6.7; *p* = 0.167		
**MTG**	Pre-treatment	0	6	7	0	11	2
	Post-treatment	4	6	3	5	7	1
	Follow-up	0	8	5	3	8	2
	χ^2^ analysis	χ^2^ (4) = 10.0, *p* = **0.042**			χ^2^ (4) =6.2; *p* = 0.199		
**CG**	Pre-treatment	0	2	10	0	9	3
	Post-treatment	1	5	6	0	10	2
	Follow-up	3	6	3	2	7	3
	χ^2^ analysis	χ^2^ (4) = 9.4, *p* = 0.052			χ^2^ (4) =4.8; *p* = 0.419		

FRG: Foam rolling group; MTG: Manual therapy group; CG: Control group; HDI: Headache disability inventory. Significant differences are highlighted in bold.

## Data Availability

The study protocol and de-identified individual participant data generated during this study are available from the investigators upon reasonable request from the publication. Requests should be directed to the corresponding author by email.

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
