# Peer review of "Effects of Foam Rolling vs. Manual Therapy in Patients with Tension-Type Headache: A Randomized Pilot Study"

_jcm, 2022, doi:10.3390/jcm11071778_

Round 1

Reviewer 1 Report

This is a well written piece of work. The methods and results are presented clearly. 

In the introduction line 63, although many drugs can be used for TTH as for any other medical disorder, there are very few evidence based treatments in TTH. 

Line 69 - reports improved disability and other symptoms such as photophobia and phonophobia. Although this may indeed be a flaw in the current International Classification for Headache, by definition TTH is not disabling. Moreover, once there is more than one feature ( and throbbing is noted as a clinical characteristic in Table 2) - these are all defining features of migraine not TTH. 

Later on the paper the HIT-6 is seen at 64. Again by definition TTH is not disabling. Thus what is the interpretation of this? 

Although pericranial tenderness remains in the classification this is synonymous with allodynia which is seen in all headache disorders especially with chronicity. 

There is only a single reference for the pathological basis of TTH and the rationale for the treatments proposed (reference 19), however this is not about TTH. 

What is the vertebral artery test and what does it indicate? 

There is no reference(s) to validate that the fourth ventricle can be compressed from externally with an intact skull Line 141

From a neurological perspective referred pain is neuroanatomically explicable. However, line 160 does not seem to make sense - what is meant here by referred pain ? 

Line 171 and 172 - please provide and ideally explain references which have formally validated that the cranial bones have a lack of mobility in a TTH population compared to a normal population and how one can hypothesise this to be related pathophysiologically. 

Please explain : effleurage, digital ptrissage and tapotement - these are not  familiar medical terms. 

Line 198 - ' interscapular muscles of the knees' Is this a typo? 

Was there any medication overuse ? Were analgesics allowed during the study? Were any patients on preventives? 

The authors have done well to include a control group which is difficult is such studies. Yet in the conclusion they have largely ignored the comparison with the control group and focused on the treatment arms only, to suggest a positive response. The comparison with controls does not seem to show a significant difference between the two groups. 

In summary the patient population is not phenotypically homogenous in regard to clinical syndrome. Medication overuse remains a key factor in the interpretation and is not mentioned. The conclusion is not based on the results. The numbers are small and the duration of the study short. No comparison with outcomes of treatments effective in TTH are discussed. This is regard to comparing the difference in cost  (manpower), time and end points. 

Author Response

REVIEWER 1:

This is a well written piece of work. The methods and results are presented clearly. In the introduction line 63, although many drugs can be used for TTH as for any other medical disorder, there are very few evidence based treatments in TTH.
We have included the suggestion of the reviewer in line 62-64 : “Although drugs are considered as the first-line treatment, non-pharmacology treatments in TTH show positive results [9]. However, more studies are necessary to strengthen the evidence of non-pharmacology treatments in TTH [10]

Line 69 - reports improved disability and other symptoms such as photophobia and phonophobia. Although this may indeed be a flaw in the current International Classification for Headache, by definition TTH is not disabling. Moreover, once there is more than one feature ( and throbbing is noted as a clinical characteristic in Table 2) - these are all defining features of migraine not TTH.
We applied the suggestion regarding disability. In line 67-70 we included this information: “We previously reported that an MT treatment for TTH based on the combination of sub-occipital soft tissue inhibition and occiput-atlas-axis manipulation improved several symptoms, including photophobia, phonophobia and pericranial tenderness [12].”

On the other hand, there has been a mistake when translating the content of the table. There were no patients presenting pressure throbbing pain. We thank the reviewer for the suggestion and have included in the table “Pressing or tightening (non-pulsating)” instead of “Pressure throbbing pain”.

All subjects were diagnosed by a specialist physician.

Later on the paper the HIT-6 is seen at 64. Again by definition TTH is not disabling. Thus what is the interpretation of this?
We understand the reviewer refers to HDI. We use this validated scale to assess the disability produced by headache in patients.

What is the vertebral artery test and what does it indicate?

Thank you very much for the suggestion, we have proceeded to include the reference: “The vertebral artery test has been widely used as one of the methods of screening the vertebro-basilar system prior to manipulative therapy[26].”

[26]. Mitchell J, Keene D, Dyson C, Harvey L, Pruvey C, Phillips R. Is cervical spine rotation, as used in the standard vertebrobasilar insufficiency test, associated with a measureable change in intracranial vertebral artery blood flow? Man Ther. 2004 Nov;9(4):220-7.

There is no reference(s) to validate that the fourth ventricle can be compressed from externally with an intact skull Line 141
In our study we have proceeded to carry out the same application of the technique as other authors with the same duration:

The compression-traction was maintained until 10 minutes [15].

[15].Hanten, W.P., Olson, S.L., Hodson, J.L., Imler, V.L., Knab, V.M. and Magee, J.L., 1999. The Effectiveness of CV-4 and Resting Position Techniques on Subjects with Tension-Type Headaches. Journal of Manual & Manipulative Therapy. 7, 64-70.

From a neurological perspective referred pain is neuroanatomically explicable. However, line 160 does not seem to make sense - what is meant here by referred pain ?
Thanks for the suggestion, we thought the description was wrong.

Force was progressively applied to the nodule, with the patient instructed to verbally indicate whether they felt pain locally or referred pain.

Line 171 and 172 - please provide and ideally explain references which have formally validated that the cranial bones have a lack of mobility in a TTH population compared to a normal population and how one can hypothesise this to be related pathophysiologically.

All of these manual osteopathic treatments are believed to improve circulation, release joint restrictions, reduce tension in muscles, fascia, and dura, decrease nociceptive

input, and promote a normalizing or calming effect on the CNS according to Hanten et al. [17]
So, we have changed the sentence to make it easier to understand: “Since the possible relationship between restrictions of the skull, sacrum, and iliac bones and CT has been reported”

Please explain: effleurage, digital petrissage and tapotement - these are not familiar medical terms.
These massage techniques are known in the area of manual therapy or physiotherapy and this is how it is described in scientific texts that study the treatments in which massage is applied:

The effleurage technique is a type of Swedish massage that is performed with a gentle, whole hand movement in contact with the body. The petrissage technique consisted of horizontal or vertical rolling and kneading techniques. Both effleurage and petrissage techniques were applied with a slow rhythm and low pressure. Tapotement, particularly ulnar hacking and tapping.

Related references:

  • -  Andrade, C., 2013. Outcome-Based Massage: Putting Evidence into Practice,

    third ed. Lippincott Williams & Wilkins, Philadelphia, PA.

  • -  Beck, M.F., 2017. Theory & Practice of Therapeutic Massage, sixth ed. Milady

    Publishing, Boston, MA.

    Line 198 - ' interscapular muscles of the knees' Is this a typo?

    We thank the reviewer for this comment, since this is a typo. We have proceeded to modify this information in the text.

    Was there any medication overuse ? Were analgesics allowed during the study? Were any patients on preventives?
    There was no medication overuse, symptomatic analgesics were allowed, and no patients were on preventives. We have added in table 1:

  • -  No medication overuse.

  • -  No medication preventive.

    The authors have done well to include a control group which is difficult is such studies. Yet in the conclusion they have largely ignored the comparison with the control group and focused on the treatment arms only, to suggest a positive response. The comparison with controls does not seem to show a significant difference between the two groups. In summary the patient population is not phenotypically homogenous in regard to clinical syndrome. Medication overuse remains a key factor in the interpretation and is not mentioned. The conclusion is not based on the results. The numbers are small and the duration of the study short. No comparison with outcomes of treatments effective in TTH are discussed. This is regard to comparing the difference in cost (manpower), time and end points.

    In line 226-227: we have included the following sentence: “Medication overuse remains a key factor in the interpretation, although in our sample there was no medication overuse”.

Reviewer 2 Report

The paper presents an interest approach to an interest topic and is really well The Introduction and Methodology are written, well documented and clearly presented. Some important references are very old, older then 40 years. Especially to define the visual analogue scale (VAS), according with new reference. 

References :

  1. Annals of the Rheumatic Diseases, 1978, 37, 378-381
    Studies with pain rating scales
    W. W. DOWNIE*, P. A. LEATHAM*, V. M. RHIND*, V. WRIGHT*,
    J. A. BRANCOt, AND J. A. ANDERSONt

    2. Annals of the Rheumatic Diseases, 1979, 38, 560
    Vertical or horizontal visual analogue scales
    JANE SCOTT AND E. C. HUSKISSON

references are old.

Better is used novelty references:

  1. Comparison of anxiety and pain perceived with conventional and computerized local anesthesia delivery systems for different stages of anesthesia delivery in maxillary and mandibular nerve blocks.

Kamal Aggarwal, Arundeep Kaur Lamba, Farrukh Faraz, Shruti Tandon, Kanika Makker

J Dent Anesth Pain Med 2018;18(6):367-373❚https://doi.org/10.17245/jdapm.2018.18.6.367

Where VAS is defined as:

The visual analogue scale (VAS) was used for the evaluation of pain [7]. VAS was scored on a 100-mm horizontal line with the left end marked “no pain” and the right end “severe intolerable pain”

  1. MEASUREMENT OF PAIN

Joel Katz, PhD, and Ronald Melzack, PhD

Surg Clin Nort Am. 1999 Apr;79(2):231-52. doi: 10.1016/s0039-6109(05)70381-9.

Section: Verbal and Numeric Rating Scales

And according new definition show the experimental results.

But, when is used, Verbal rating scales typically consist of a series of verbal pain descriptors ordered from least to most intense (e.g., no pain, mild,
moderate, or severe). A score of 0 is assigned to the descriptor with the lowest rank, a score of 1 is assigned to the descriptor with the next lowest rank, and so on.    

The visual analogue scale and The Verbal rating scales are very often mixed in researches and presented in very high IF journal, therefor the editor and authors will make decision about correction representacion of the Visual analogue scale (VAS) and Verbal rating scales. VAS is analogue scale, in my opinion, depicted in mm :-)      

The paper is well formatted with all sections to be detailed and in the right order.

Author Response

REVIEWER 2:

The paper presents an interest approach to an interest topic and is really well The Introduction and Methodology are written, well documented and clearly presented. Some important references are very old, older than 40 years. Especially, to define the visual analogue scale (VAS), according with new reference. References :

1. Annals of the Rheumatic Diseases, 1978, 37, 378-381 Studies with pain rating scales W. W. DOWNIE*, P. A. LEATHAM*, V. M. RHIND*, V. WRIGHT*, J. A. BRANCOt, AND J. A. ANDERSON
2. Annals of the Rheumatic Diseases, 1979, 38, 560. Vertical or horizontal visual analogue scales JANE SCOTT AND E. C. HUSKISSON

references are old.

We thank the reviewer for the suggestion and have deleted the reference from the manuscript.

Better is used novelty references:
1. Comparison of anxiety and pain perceived with conventional and computerized local anesthesia delivery systems for different stages of anesthesia delivery in maxillary and mandibular nerve blocks. Kamal Aggarwal, Arundeep Kaur Lamba, Farrukh Faraz, Shruti Tandon, Kanika Makker J Dent Anesth Pain Med 2018;18(6):367373❚https://doi.org/10.17245/jdapm.2018.18.6.367
We have included this reference in the manuscript :
[34]. Aggarwal K, Lamba AK, Faraz F, Tandon S, Makker K. Comparison of anxiety and pain perceived with conventional and computerized local anesthesia delivery systems for different stages of anesthesia delivery in maxillary and mandibular nerve blocks. J Dent Anesth Pain Med. 2018 Dec;18(6):367-373.
Where VAS is defined as:

The visual analogue scale (VAS) was used for the evaluation of pain [7]. VAS was scored on a 100-mm horizontal line with the left end marked “no pain” and the right end “severe intolerable pain”

2. MEASUREMENT OF PAIN
Joel Katz, PhD, and Ronald Melzack, PhD Surg Clin Nort Am. 1999 Apr;79(2):231-52. doi: 10.1016/s0039-6109(05)70381-9.
Katz J, Melzack R. Measurement of pain. Surg Clin North Am. 1999 Apr;79(2):231-52. Section: Verbal and Numeric Rating Scales
And according new definition show the experimental results.
But, when is used, Verbal rating scales typically consist of a series of verbal pain descriptors ordered from least to most intense (e.g., no pain, mild, moderate, or severe). A score of 0 is assigned to the descriptor with the lowest rank, a score of 1 is assigned to the descriptor with the next lowest rank, and so on.
The visual analogue scale and The Verbal rating scales are very often mixed in research and presented in very high IF journal, therefor the editor and authors will make decision about correction representation of the Visual analogue scale (VAS) and Verbal rating scales. VAS is analogue scale, in my opinion, depicted in mm :-)
The paper is well formatted with all sections to be detailed and in the right order.

We have proceeded to put a more updated reference according to your suggestion.

Reviewer 3 Report

This study compares the effect of Foam Rolling (FR) vs Manual therapy (MT) 17 on pain, pressure pain threshold (PPT), headache disability (HDI) and impact of headache (HIT-6) 18 in patients with tension-type headache (TTH), concluding that Both FR and MT are effective whereas further studies are needed to confirm these findings in a larger population. There are some minor concerns:

Lines 131 -213. Did the patients communicate with each other the different durations of treatments (40, 25 and 10 minutes)? This aspect is important to try to reduce the variability between groups.

Lines 303-304: There are differences between the groups regarding the duration of the disease and the frequency of attacks. This aspect needs to be discussed.

Table 4 needs to be shortened and simplified to increase understanding.

Line 400  “For CG, pre-post significant differences were found in emotional, functional and overall 400 HDI and HIT-6 score,..” This is an interesting point and should be discussed.

Line 435: The duration of the effect was measured in a short time, and this point should be discussed in relation to the placebo effect found in similar situations, such reported in line 485.

Author Response

REVIEWER 3:

This study compares the effect of Foam Rolling (FR) vs Manual therapy (MT) 17 on pain, pressure pain threshold (PPT), headache disability (HDI) and impact of headache (HIT-6) 18 in patients with tension-type headache (TTH), concluding that Both FR and MT are effective whereas further studies are needed to confirm these findings in a larger population. There are some minor concerns:

Lines 131 -213. Did the patients communicate with each other the different durations of treatments (40, 25 and 10 minutes)? This aspect is important to try to reduce the variability between groups.

The informed consent included information regarding the differences between groups, including differences between treatments and duration.

Lines 303-304: There are differences between the groups regarding the duration of the disease and the frequency of attacks. This aspect needs to be discussed.

No significant differences were found in any of the descriptive variables shown in Table 2. Duration of the disease (Evolution of headache (years)) presents a p=0.06. Frequency of attacks showed a p=0.99 and severity of disorder a p=0.06. We have considered significant the differences with a p<0.05.

Differences in the duration of the disease and the frequency of attacks may be due to the randomization of the sample. We have added the reviewer suggestion in the discussion, line 369-373: “We found differences in the evolution of headache in years, and in the severity of the disorder. The control group presented more years of evolution of headache and more severe headaches than FRG and MTG. Nevertheless, these differences may be due to the randomization of the sample.”

Table 4 needs to be shortened and simplified to increase understanding.

Following your suggestion, we have improved the format and simplified the table to improve its understanding.

Line 400 “For CG, pre-post significant differences were found in emotional, functional and overall 400 HDI and HIT- 6 score,..” This is an interesting point and should be discussed.

Lines 354-358: We are include the following discussion text: “ We have observed an effect of the placebo technique in some of the variables studied, specially referred to pre-post significant differences observed in emotional, functional and overall HDI and HIT- 6 score for CG, which it might point to placebo effect. We observe that globally, these improvements were smaller (smaller effect size) than in the FRG and MTG.”

Line 435: The duration of the effect was measured in a short time, and this point should be discussed in relation to the placebo effect found in similar situations, such reported in line 485.

Lines 358-361: We are include the following discussion text: “This placebo effect that we have observed could be due to the fact that the follow-up was carried out in a very limited period of time (one month), so we think that if the follow-up period had been longer (two or three months), perhaps we would not have observed said placebo effect, which should be considered in future research.”

Reviewer 4 Report

This will be a useful contribution to the headache field. I have the following concerns.

  1. Please use CONSORT for your manuscript to fit to standards, and provide CONSORT checklist.
  2. Use mixed models for repeated measures for analysis of results instead of ANOVA. You have repeated measures, etc, hence ANOVA is not applicable. 
  3. Report missing data and if it exists, how it was handled.
  4. Since you have several outcome measures, clarify which is the primary, secondary, etc outcomes. Also, include confounder, moderator, and mediator analysis in your study for the different baseline and outcome variables displayed.
  5. I find the sample size very low. The justification for the very high effect size needs to be cited. This study needs to be presented as a preliminary or pilot result, given the very low sample size in a highly heterogeneous and very common condition like tension-type headache.

Author Response

REVIEWER 4:

This will be a useful contribution to the headache field. I have the following concerns.

1. Please use CONSORT for your manuscript to fit to standards, and provide CONSORT checklist.

We have included the Consort Checklist in the new submission.

2. Use mixed models for repeated measures for analysis of results instead of ANOVA. You have repeated measures, etc, hence ANOVA is not applicable.

As explained in section 2.7 (Statistical analysis), we have used one-factor ANOVA to compare the participants’ characteristics (demographic and clinical data). However, to analyze the outcome measures, we have used a mixed 2-factor ANOVA model with repeated measures:

Line 265: “A mixed 2-factor ANOVA with repeated measures in the time factor was used to determine significant differences in the measured variables between time points (pre-, post-treatment and follow-up) and groups (FRG, MTG and CG).”

We think that the used model is perfectly valid for the analysis performed.

3. Report missing data and if it exists, how it was handled.

We confirm that there are no missing data in any of the analyzed variables.

Line 286: We are including the following text: “None of the analyzed variables presented missing data.”

4. Since you have several outcome measures, clarify which is the primary, secondary, etc outcomes. Also, include confounder, moderator, and mediator analysis in your study for the different baseline and outcome variables displayed.

In the methods section there is a subsection dedicated to the outcome variables (2.5 Outcome measures). In subsections 2.5.1 and 2.5.2 the primary and secondary variables are described, respectively.

Since there are no differences between groups in the clinical and demographic variables, we think that it is not necessary to use these variables as covariates or confounders in the

outcome variables. In addition, we think that performing an exhaustive moderator and mediator analysis is beyond the scope of this work.

5. I find the sample size very low. The justification for the very high effect size needs to be cited. This study needs to be presented as a preliminary or pilot result, given the very low sample size in a highly heterogeneous and very common condition like tension-type headache.

We greatly appreciate your suggestion. Although there are numerous published studies with a small sample, which has been estimated based on the sample size calculation, we agree with you, and for this reason we have made the changes for a pilot study design. Despite this, we would like to maintain the data on the calculation of the sample size, even though our study is now a pilot study, since in this way our study would have greater methodological rigor as a guide for future research derived from our research. We think that our research would be methodologically more rigorous by providing the estimated data from the sample calculation, although ultimately our research is a pilot study, assuming your suggestions.

We have included the following changes in the text, related to the pilot study: In the study title, the study design section, and discussion section.

New study title: “Effects of foam rolling versus manual therapy in patients with tension- type headache: A randomized pilot study.”

Line 116: 2.1 Section - Study Design: “ The study was a prospective, assessor-blinded, randomized pilot study. Level of Evidence: Therapy, level 2.”

Line 471-472 : Discussion – 4.1 Strengths and limitations: “However, this study has limitations, such as the small sample size, although as a pilot study it could be sufficient according to a priori power analysis.”

Round 2

Reviewer 1 Report

I still find too many flaws in the paper particularly in the nature of the basic  methodology in regards to headache, and hence the interpretations. 

Author Response

In response to your suggestions, we have considered the following modifications in the text of the manuscript, additionally increasing the bibliographical references:

LINES 34-36

Tension-type headache (TTH) is a primary headache reclassified by the International Headache Society (IHS) in 2018, which describes in detail the diagnostic criteria for each type of headache.

LINES 39-42

The increased tenderness in the pericranial muscles is the most significant abnormal finding seen in patients with TTH. This tenderness typically occurs between attacks, worsens during the headache, and increases with intensity and frequency. Hypersensitivity seems to have a pathophysiologic import. [1]

LINES 56-65

This type of headache was considered primarily psychogenic, however, several studies strongly suggested the possibility of a neurobiological basis, at least for the most severe subtypes of TTH. The differences between episodic and chronic TTH that were presented in ICHD-I have been proven to be extremely useful to clinicians, especially in those cases where diagnosis is uncertain[1]. The IHS recommends to compare patients diagnosed according to each group of criteria, not only to categorize the clinical characteristics, but also to investigate the pathophysiological mechanisms and the response to treatments [1]. Several factors can trigger headache, such as emotional and genetic factors, as well as peripheral factors (pericranial tenderness, muscle strain, muscle blood flow and ischemia) or central factors such as pain sensitivity, central pain modulation, etc.

NEW REFERENCES:

 DOI: 10.1007/s11916-005-0021-8

Chen, Yaniv. "Advances in the pathophysiology of tension-type headache: from stress to central sensitization." Current pain and headache reports 13.6 (2009): 484-494. PMID: 19889292 DOI: 10.1007/s11916-009-0078-x

Reviewer 4 Report

Most of my comments are addressed satisfactorily. The only point I am not convinced is on the use of ANOVA or mixed 2-factor ANOVA for analyzing the results of a randomized clinical trial. Please redo the analysis using MMRM and demonstrate if the MMRM results are similar to the ones from ANOVA. 

Author Response

Following your suggestion, we have analyzed the data using MMRM models. The base code used in SPSS is as follows:

MIXED Pain BY Time Group

  /CRITERIA=DFMETHOD(SATTERTHWAITE) CIN(95) MXITER(100) MXSTEP(10) SCORING(1)     SINGULAR(0.000000000001) HCONVERGE(0, ABSOLUTE) LCONVERGE(0, ABSOLUTE) PCONVERGE(0.000001, ABSOLUTE)   

  /FIXED=Time Group Time*Group | SSTYPE(3)

  /METHOD=REML

/RANDOM=INTERCEPT | SUBJECT(ID) COVTYPE(ID)

  /PRINT=CORB COVB DESCRIPTIVES  SOLUTION TESTCOV

  /REPEATED=Time | SUBJECT(ID) COVTYPE(CS)

 /EMMEANS=TABLES(Group*Time) COMPARE(Group) ADJ(BONFERRONI)

  /EMMEANS=TABLES(Time*Group) COMPARE(Time) ADJ(BONFERRONI)

From this code we have tried different types of covariance matrices and different random factors. To check which model best fits the data, we have tried to minimize the Resticted Log Likelihood and the AIC information criteria. The following table shows the results of the information criteria for the different covariance matrices for the pain variable.

Information Criteriaa

Repeated covariance type

Compound Symmetry

Unstructured

First-Order Autoregressive

Diagonal

-2 Restricted Log Likelihood

370,160

362,251

372,978

383,658

Akaike's Information Criterion (AIC)

374,160

374,251

376,978

389,658

The information criteria are displayed in smaller-is-better form.

a. Dependent Variable: Pain.

As can be seen, with the unstructured covariance type, the lowest values are obtained. The intercept as well as the intercept, group and time were then added as random factors to the model. Using these random factors did not improve the model.

Information Criteriaa

Repeated covariance type Unstructured

Random Effects Intercept

Random Effects Intercept + Group+ Time

-2 Restricted Log Likelihood

362,251

362,251

Akaike's Information Criterion (AIC)

376,251

380,251

The information criteria are displayed in smaller-is-better form.

a. Dependent Variable: Pain.

With the best model, the results obtained were the following:

Type III Tests of Fixed Effectsa

Source

Numerator df

Denominator df

F

Sig.

Intercept

1

35

1090,414

,000

Time

2

35,000

8,146

,001

Group

2

35

2,799

,075

Time * Group

4

35,000

3,291

,022

a. Dependent Variable: Pain.

These results are in line with those obtained in the main effects ANOVA (Table 3 of the manuscript). In addition, the results of the pairwise comparisons are identical to those shown in Table 4.

Pairwise Comparisonsa

Time

(I) Group

(J) Group

Mean Difference (I-J)

Std. Error

df

Sig.c

95% Confidence Interval for Differencec

Lower Bound

Upper Bound

Pre

FRG

MTG

,462

,473

35

1,000

-,728

1,651

CG

,468

,483

35

1,000

-,746

1,682

MTG

FRG

-,462

,473

35

1,000

-1,651

,728

CG

,006

,483

35

1,000

-1,208

1,221

CG

FRG

-,468

,483

35

1,000

-1,682

,746

MTG

-,006

,483

35

1,000

-1,221

1,208

Post

FRG

MTG

,692

,491

35

,501

-,541

1,926

CG

-,532

,501

35

,886

-1,791

,727

MTG

FRG

-,692

,491

35

,501

-1,926

,541

CG

-1,224

,501

35

,059

-2,484

,035

CG

FRG

,532

,501

35

,886

-,727

1,791

MTG

1,224

,501

35

,059

-,035

2,484

Follow-up

FRG

MTG

,923

,638

35

,470

-,681

2,527

CG

-,833

,651

35

,627

-2,471

,804

MTG

FRG

-,923

,638

35

,470

-2,527

,681

CG

-1,756*

,651

35

,032

-3,394

-,119

CG

FRG

,833

,651

35

,627

-,804

2,471

MTG

1,756*

,651

35

,032

,119

3,394

Based on estimated marginal means

*. The mean difference is significant at the ,05 level.

a. Dependent Variable: Pain.

c. Adjustment for multiple comparisons: Bonferroni.

Pairwise Comparisonsa

Group

(I) Time

(J) Time

Mean Difference (I-J)

Std. Error

df

Sig.c

95% Confidence Interval for Differencec

Lower Bound

Upper Bound

FRG

Pre

Post

1,000*

,319

35,000

,010

,198

1,802

Follow-up

,385

,429

35,000

1,000

-,694

1,463

Post

Pre

-1,000*

,319

35,000

,010

-1,802

-,198

Follow-up

-,615

,462

35,000

,574

-1,777

,546

Follow-up

Pre

-,385

,429

35,000

1,000

-1,463

,694

Post

,615

,462

35,000

,574

-,546

1,777

MTG

Pre

Post

1,231*

,319

35,000

,001

,429

2,033

Follow-up

,846

,429

35,000

,169

-,232

1,925

Post

Pre

-1,231*

,319

35,000

,001

-2,033

-,429

Follow-up

-,385

,462

35,000

1,000

-1,546

,777

Follow-up

Pre

-,846

,429

35,000

,169

-1,925

,232

Post

,385

,462

35,000

1,000

-,777

1,546

CG

Pre

Post

1,044E-14

,332

35,000

1,000

-,835

,835

Follow-up

-,917

,446

35,000

,143

-2,039

,206

Post

Pre

-1,044E-14

,332

35,000

1,000

-,835

,835

Follow-up

-,917

,481

35,000

,194

-2,126

,292

Follow-up

Pre

,917

,446

35,000

,143

-,206

2,039

Post

,917

,481

35,000

,194

-,292

2,126

Based on estimated marginal means

*. The mean difference is significant at the ,05 level.

a. Dependent Variable: Pain.

c. Adjustment for multiple comparisons: Bonferroni.

We have also tried to add age, sex, bmi, evolution of headache or severity of disorder as covariates. None of the added variables showed significance.

Type III Tests of Fixed Effectsa

Source

Sig.

Intercept

,502

Time

,001

Group

,124

Time * Group

,027

Sex

,512

Age

,346

BMI

,843

Evolution of headache

,317

Severity of disorder

,105

a. Dependent Variable: Pain.

Following the same strategy, the rest of the variables have been analyzed using MMRM models. The results are shown in the following tables. In all cases, the best model is obtained with the unstructured covariance matrix and without random factors. Results obtained in the fixed effects are similar to those obtained in the ANOVA models.

Information Criteriaa

Repeated covariance type

Compound Symmetry

Unstructured

First-Order Autoregressive

Unstructured and

With intercep random effect

-2 Restricted Log Likelihood

162,637

151,398

171,351

151,398

Akaike's Information Criterion (AIC)

166,637

163,398

175,351

165,398

The information criteria are displayed in smaller-is-better form.

a. Dependent Variable: PptRightSub.

Type III Tests of Fixed Effectsa

Source

Numerator df

Denominator df

F

Sig.

Intercept

1

35

514,208

,000

Time

2

35,000

5,412

,009

Group

2

35

1,246

,300

Time * Group

4

35,000

7,258

,000

a. Dependent Variable: PptRightSub.

Information Criteriaa

Repeated covariance type

Compound Symmetry

Unstructured

First-Order Autoregressive

Unstructured and

With intercep random effect

-2 Restricted Log Likelihood

128,489

122,029

140,817

122,029

Akaike's Information Criterion (AIC)

132,489

134,029

144,817

136,029

The information criteria are displayed in smaller-is-better form.

a. Dependent Variable: PptLeftSub.

Type III Tests of Fixed Effectsa

Source

Numerator df

Denominator df

F

Sig.

Intercept

1

35

672,259

,000

Time

2

35

3,444

,043

Group

2

35

2,504

,096

Time * Group

4

35

4,143

,008

a. Dependent Variable: PptLeftSub.

Information Criteriaa

Repeated covariance type

Compound Symmetry

Unstructured

First-Order Autoregressive

Unstructured and

With intercep random effect

-2 Restricted Log Likelihood

654,365

634,855

655,435

634,855

Akaike's Information Criterion (AIC)

658,365

646,855

659,435

648,855

The information criteria are displayed in smaller-is-better form.

a. Dependent Variable: HIT6.

Type III Tests of Fixed Effectsa

Source

Numerator df

Denominator df

F

Sig.

Intercept

1

35,000

5704,533

,000

Time

2

35,000

49,208

,000

Group

2

35,000

1,725

,193

Time * Group

4

35,000

3,792

,012

a. Dependent Variable: HIT6.

Information Criteriaa

Repeated covariance type

Compound Symmetry

Unstructured

First-Order Autoregressive

Unstructured and

With intercep random effect

-2 Restricted Log Likelihood

876,195

866,388

873,330

866,388

Akaike's Information Criterion (AIC)

880,195

878,388

877,330

880,388

The information criteria are displayed in smaller-is-better form.

a. Dependent Variable: HDI total.

Type III Tests of Fixed Effectsa

Source

Numerator df

Denominator df

F

Sig.

Intercept

1

35

208,164

,000

Time

2

35,000

34,878

,000

Group

2

35

1,749

,189

Time * Group

4

35,000

,854

,501

a. Dependent Variable: HDItotal.

We think that the statistical analysis of the manuscript is adequate since the results are similar to the new statistical analysis that you have requested. However, if you consider it necessary, we could replace the statistical analysis, so we would change it in the manuscript for the new statistical analysis if you request it.
